# Lipid Metabolism and Endocrine Resistance in Prostate Cancer, and New Opportunities for Therapy

**DOI:** 10.3390/ijms20112626

**Published:** 2019-05-28

**Authors:** Gergana E. Stoykova, Isabel R. Schlaepfer

**Affiliations:** Division of Medical Oncology, University of Colorado School of Medicine, Genitourinary Cancer Program, MS 8117, 12801 E. 17th Ave, Room L18-8119, Aurora, CO 80045, USA; gergana.stoykova@ucdenver.edu

**Keywords:** lipid synthesis, lipid oxidation, AR, prostate cancer, CPT1A, FASN, endocrine resistance, anti-androgens, dietary lipids, combination therapy

## Abstract

Prostate cancer (PCa) is the most common cancer in men, and more than 10% of men will be diagnosed with PCa during their lifetime. Patients that are not cured with surgery or radiation are largely treated with endocrine therapies that target androgens or the androgen receptor (AR), a major driver of PCa. In response to androgen deprivation, most PCas progress to castrate resistant PCa, which is treated with anti-androgens like enzalutamide, but tumors still progress and become incurable. Thus, there is a critical need to identify cellular pathways that allow tumors to escape anti-androgen therapies. Epidemiological studies suggest that high-fat diets play important roles in PCa progression. Lipid metabolism rewires the PCa metabolome to support growth and resistance to endocrine therapies, although the exact mechanisms remain obscure. Therapeutic effects have been observed inhibiting several aspects of PCa lipid metabolism: Synthesis, uptake, and oxidation. Since AR remains a driver of PCa in advanced disease, strategies targeting both lipid metabolism and AR are starting to emerge, providing new opportunities to re-sensitize tumors to endocrine therapies with lipid metabolic approaches.

## 1. Introduction

Prostate cancer (PCa) is the most common cancer in men and the second highest contributor to cancer deaths in the Western hemisphere [1]. For men diagnosed with localized PCa, the main options include active surveillance, surgery, and radiation. Both surgery and radiation have undergone significant technological advances like robotic surgery and 3D-confromal radiation [2]. For advanced PCa, the first line of treatment is based on androgen deprivation therapy (ADT). This testosterone lowering approach is based on the work of Huggins in decades past [3]. Most patients respond favorably to ADT, however, resistance to ADT develops overtime and patients become insensitive to the androgen withdrawal or androgen receptor (AR) inhibition [4], limiting the options for treatment significantly. Preclinical models have identified mechanisms by which androgen activity is maintained in the presence of ADT, which include increased androgen synthesis, AR gene amplification, AR mutations in the ligand-binding domain, and AR alternative splice variants [5].

From the metabolic perspective, men treated with ADT experience adverse effects such as fatigue, sarcopenia, loss of bone density, increased adiposity, and metabolic syndrome [6,7]. Additionally, ADT causes an increase in total cholesterol, triglycerides, and high-density lipoproteins [8,9]. Thus, ADT is significantly associated with metabolic and lipid-specific changes in PCa patients. The role and relative contribution of lipid and glucose metabolism in PCa is different from other cancers. Although most cancers mainly use glycolysis, recent evidence suggests that PCa prefers to use lipid metabolism instead. In fact, extensive research is focused on the identification of lipid synthesis enzymes as therapeutic targets, but finding safe drugs remains challenging. [10,11]. The overexpression of key enzymes of lipid metabolism in PCa is characteristic of both primary and advanced disease [12], suggesting that targeting lipid enzymes in PCa may be more relevant than in other cancer types.

This review will cover three emerging, conceptually related aspects of PCa research: Lipid metabolism, endocrine therapy challenges, and potential therapeutic opportunities targeting both lipid and androgen signaling.

## 2. Prostate Cancer and Lipid Metabolism

The observation of a close correlation between the average per capita fat intake and PCa mortality in several countries, including the USA, sparked a lot of interest in the potential link between dietary fat and prostate cancer risk [13]. It has been widely postulated that a Western diet can promote PCa progression to lethal disease [14]. However, direct evidence supporting a strong association between specific dietary lipids and PCa is still obscure. Studies in animal models have shown increased tumor growth with high fat intake and inhibition of growth with low fat intake. Other studies have also shown that reduced dietary fat intake in xenograft-bearing nude mice delays the progression of PCa to androgen insensitivity and prolongs survival [15]. On the other hand, there are other preclinical studies that have found no relationship between the growth of transplanted PCa and variations in dietary fat [16]. Several studies have made correlative associations between dietary lipid composition and PCa (Table 1). The impact of dietary fat on the development and progression of PCa remains unknown and controversial, as the selection of populations and dietary interventions vary widely among studies. However, all these correlative studies strongly suggest that lipid availability to the cancer cells, whether newly synthesized or exogenously acquired, likely promotes PCa growth and progression.

### 2.1. Lipid Synthesis and Uptake

Lipid metabolism in cancer can be studied from two angles: Lipid synthesis and lipid breakdown, which is then likely used for oxidation and ATP synthesis. The former has received more attention in the past due to the identification of de novo lipogenesis as a biomarker of aggressive disease in many hormone-dependent cancers [17]. In fact, androgens are known to modulate lipid synthesis in PCa cells [10,18]. The enzyme fatty acid synthase (FASN) is the first step for de novo fatty acid synthesis in cancer cells [10,19]. In the last decade, intense research has been focused on designing or re-purposing FASN inhibitors to block the ability of cancer cells to make their own lipids [20,21]. In fact, a recent study using genetic models of PCa has convincingly shown that a high-fat diet promotes metastatic prostate cancer, likely mediated by an aberrant lipogenic program orchestrated by the transcription factor SREBP [22]. An interesting aspect of this study is that the genetic component did not matter as much when the mice were fed a high-fat-containing diet. The saturated lipid content of the diet (>60%) was enough to promote aggressive metastatic PCa in the mice. These results speak to the fact that lipids inside the cancer cells may provide an intrinsic signal that promotes aggressive PCa subtypes [23]. Although this genetic rodent study represents a great step forward in understanding how fat promotes metastatic PCa, it does not address other critical questions: How do the tumor cells use the fat? Is there a preference for long chain fatty acids, the most common fatty acids circulating in blood? Does androgen signaling in these mouse tumors play a role? These questions remain unsolved.

Another important aspect of lipid metabolism beyond de novo lipid synthesis is the ability of many tumors to desaturate their own lipids. This desaturation of newly synthesized fatty acids is necessary for the formation of lipid droplets and membranes, and promotes growth and survival of the cancer cells [24]. Particularly, the enzyme stearoyl CoA desaturase-1, or SCD-1, has been shown to be induced by progestins in breast cancer cells [25]. This SCD-1 upregulation was associated with increased viability, intracellular lipid droplet accumulation, and resistance to docetaxel, which was sequestered in the lipid droplets. This accumulation of an active drug in hormone-induced lipid droplets suggests potential new roles for lipid synthesis in hormone-dependent cancers. Since androgens induce the accumulation of lipid droplets as well [18,26], it is very possible that chemotherapeutic or endocrine agents used in PCa would accumulate in lipid droplets and support therapy resistance. More recently, a new pathway of desaturation in PCa cells and other cancers has been identified. Researchers in Belgium have identified fatty acid desaturase 2, or FADS2, as an alternative desaturase pathway in tumors [24]. Cancer cells use this pathway when SCD-1 is inhibited, underscoring the metabolic plasticity of cancer cells for making their own lipids for growth and resistance to therapy.

Cancer cells can also obtain fatty acids from the circulation, by breaking down triglycerides contained in circulating chylomicrons and lipoproteins. In fact, several studies have found significant associations between circulating lipoproteins and PCa growth and recurrence (Table 2). However, our understanding of the lipid delivery to tumors via lipoproteins or exosomes remains obscure. Evidence is also pointing to the role of fibroblasts surrounding tumors as suppliers of nutrients and growth signals. In fact, it has been proposed that cancer-associated fibroblasts are forced to produce energy-rich metabolites (like ketones from lipids) to feed adjacent cancer cells [40]. Other studies have shown that loss of AR in the stroma also supports the progression of PCa by promoting resistance to anti-androgen therapies [41], although the role of lipids in this process is unknown. Newer studies using PCa organoids and lipids will help elucidate how the tumor microenvironment promotes PCa growth and progression [42,43]. The current understanding of lipid supply from uptake of exogenous lipids and its regulation by AR is limited, and exogenous lipids may play a much more significant role in PCa progression than previously thought. In fact, androgens modulate the expression of multiple lipid transporters in PCa cells and tumors, and the expression of lipid transporters is enhanced in bone metastasis when compared to localized PCa disease [44]. Tousinant et al. examined 42 lipid transporter genes and highlighted the importance of the exogenous fatty acid supply to PCa tumors. Although redundancy in all these transporters is expected, it likely provides PCa tumors with the ability to switch between de novo lipogenesis and lipid uptake, facilitating adaptation to the energetic demands of disease progression and therapy resistance. Other recent studies have shown that suppressing fatty acid uptake in PCa has a significant therapeutic effect [42]. In this study, they used tissue from patients with PCa and patient-derived xenograft (PDX) mouse models, and identified the fatty acid translocase (FAT)/CD36, [45] as a key fatty acid transporter associated with aggressive disease. The authors also targeted de novo lipogenesis in combination with CD36 blockade and observed an enhanced therapeutic effect in PDX-derived human organoids. When the authors analyzed the lipidomic profile of the mice doubly deficient in CD36 and PTEN (phosphatase and tensing homolog) compared to mice only deficient in PTEN, a specific set of lipids only modulated by the CD36 transporter emerged. These included monoacylglycerols, lysophospholipids, and acyl-carnitines, suggesting that fatty acid transport can modulate the lipid signaling and fatty acid oxidation capabilities of the PCa tumors.

These results also bring attention to the fact that lipid metabolism in PCa is quite dynamic, switching between lipid synthesis and oxidation depending on the cellular context. For example, as tumors start to outgrow the vasculature network, accessibility to nutrients becomes compromised. Cells distant from blood vessels have diminished access to nutrients and oxygen, and may switch to alternative forms of metabolism like oxidation of branched amino acids and fatty acids to support viability [46]. In fact, hypoxic conditions have been shown to make PCa cells more dependent on fatty acid oxidation for survival and resistance to radiation [47]. As the tumors fragment or disseminate, re-exposure to oxygen and nutrients may induce a switch back to glycolysis and de novo lipid synthesis metabolism, preparing the tumors for future environmental insults. That is, a cycle of fat synthesis and degradation seems to promote tumor survival. In fact, targeting fat synthesis and oxidation simultaneously has been shown to reduce PCa viability, as it can strongly reduce AR expression [20]. Further understanding on how androgens may modulate this lipid synthesis/oxidation metabolic switching will aide in the design of more targeted therapies.

### 2.2. Lipid Breakdown and Oxidation

Regarding the other side of the coin, lipid utilization, significant findings have been uncovered in recent years. A key player in lipid breakdown is the rate-limiting enzyme palmitoyltransferase 1 (CPT1). For the long chain fatty acids like palmitate to get into the mitochondria, the fatty acids (acyl-CoA chains) must be enzymatically converted to acyl-carnitines across the outer mitochondrial membrane by CPT1 (Figure 1). This is a rate-limiting step for fat oxidation in both healthy and cancer cells. Although human studies looking at lipid oxidation in the setting of cancer are rare, several studies have focused on human populations carrying a mutation in the *CPT1A* (liver isoform) gene that confers some protection from PCa risk to arctic populations [48]. In fact, methylation status in the *CPT1A* gene locus significantly correlates with very low density lipoprotein (VLDL) and low density lipoprotein (LDL) lipoprotein profiles [49], pointing to an epigenetic role of this gene in metabolic dysfunction and cancer.

Studies focused on the role of lipid oxidation via CPT1A have brought attention to the role of lipid catabolism by cancer cells. [20,62,63]. Recent studies have shown that CPT1A is strongly expressed in several cancers, including the hormone-dependent breast and prostate cancers [64]. These observations suggest that lipid catabolism is a key player in the plasticity of cancer metabolism, and likely aids the cancer cells to adapt and survive harsh environments like starvation, hypoxia, and anoikis [65,66,67]. Several lines of evidence indicate that intracellular lipid oxidation is important in cancer cell survival [68], resistance to radiation [69], oxidative stress [70], and more recently, resistance to anoikis [71], activation of oncogenic signaling pathways [72], and anti-androgen resistance [73]. Altogether, lipid oxidation is an important component of metabolic reprograming in cancer that remains to be exploited for therapy in hormone-dependent cancers.

A critical link between cancer lipid metabolism and targeting it therapeutically is identifying the upstream regulators that modulate tumor metabolism. The proto-oncogene *MYC* influences multiple seemingly unrelated phenotypes, and it is commonly amplified or overexpressed in human cancers [74]. MYC is a transcription factor that de-regulates a wide variety of processes including proliferation, apoptosis, and metabolism, supporting cancer growth. In breast cancers, MYC is increased in the estrogen, progesterone, and human epidermal growth factor receptor-2 (HER2) receptor triple-negative subtype of breast cancer, or TNBC [75]. Camarda et al. found that targeting the carnitine shuttle in TNBC cells with high MYC expression resulted in decreased growth. To decrease fat oxidation, they used a pharmacological approach with etomoxir (CPT1 inhibitor) and a genetic approach via *CPT1B* knockdown. Their results suggested that fat oxidation via CPT1 was an essential metabolic pathway in MYC-overexpressing TNBC cells. The oncogenic role of MYC has also been studied in the context of transgenic mouse models of PCa, where combined MYC activation and PTEN loss synergized to induce genomic instability and aggressive PCa [76]. The role of MYC in prostate cancer fat oxidation has been less explored. MYC is known to induce aerobic glycolysis in certain preclinical models, but this was not replicated in transgenic mouse models with high MYC expression, suggesting that MYC-driven prostate cancer tumors may rely more on lipid metabolism, albeit with great heterogeneity [77]. Additional studies remain to be done on the potential roles of oncogenes like *MYC* and *AKT1* in PCa lipid utilization.

Given the wide important role of MYC in cancer, a direct MYC inhibitor could be clinically valuable. However, direct targeting of MYC remains challenging, and no inhibitor has been identified yet. Thus, targeting metabolic pathways modulated by MYC, like fat oxidation, is an intriguing therapeutic opportunity for MYC-driven cancers. Active research in this area is currently ongoing for breast and prostate cancers [78].

## 3. Endocrine Therapy Challenges in Prostate Cancer

Androgen deprivation therapy (ADT) is the standard-of-care therapy for PCa patients experiencing relapse or metastatic disease. Castrate-sensitive PCa is characterized by non-castrate levels of testosterone in the blood. This hormone-sensitive state spans from patients with increases in the prostate-specific antigen (PSA) to those with cancer metastasis detected by imaging [79]. The current clinical challenge is that virtually all patients treated with ADT eventually progress to castration-resistant prostate cancer (CRPC). This resistant state is defined as having radiographic tumor progression, with or without rising PSA, despite having castrate levels of testosterone. After the development of metastatic CRPC, median survival ranges from approximately 18 months to three years. Currently, there is no cure for CRPC, and the possibility of metabolic interventions to enhance or synergize with current endocrine therapies is compelling and worth pursuing.

A salient characteristic of CRPC is the reactivation of AR signaling. This is reflected by progressive rises in serum PSA, which is transcriptionally regulated by the AR. Ample research has shown that the majority of AR-regulated genes (androgen-response hallmark genes) are re-expressed in most CRPCs, and several mechanisms capable of maintaining AR activity have been established [80,81,82]. Due to the important role of AR signaling in advanced PCa, efforts have been directed to develop drugs that suppress AR ligands or the AR itself. Several drugs, including improved AR antagonists (enzalutamide) and inhibitors of androgen synthesis (abiraterone), extend survival [83,84], although to date, complete remissions have been rare. A consequent problem of using these potent anti-AR drugs is that they impose a strong selective pressure that pushes cancers to find alternative mechanisms distinct from those regulated by AR, or that substitute for vital AR functions. This leads to emergence of AR-null or neuroendocrine-like PCas that are very difficult to treat due to the lack of endocrine targets left in the tumor, and chemotherapy becomes the last alternative to help the patient [79,85]. Other types of selective AR drugs are also starting to emerge, like photosensitizer AR-conjugates that increase PCa cell death in response to white light [86], although these novel modalities need to be tested in humans.

Alternative splicing of AR transcripts is one of the main mechanisms implicated in the progression of CRPC as well as in resistance to anti-androgens. To date, more than 20 AR splice variants (AR-V) have been identified in prostate cancer cell lines and mouse xenografts, some of which have been validated in human cancer [81]. These AR variants have a truncated ligand-binding domain, while retaining the amino-terminal transactivation domain and DNA-binding domain of the AR protein. The best characterized variant is the AR-V7 variant, which is increased in patients resistant to the anti-androgen enzalutamide [87]. Several studies have shown that AR-V7 has unique functions in metabolic rewiring, like increased dependence on glutaminolysis, to support CRPC progression [12]. Although AR-V7 expression is clearly increased in anti-androgen-resistant tumors, the evidence that it plays a causal role in resistance remains controversial. For example, knockdown of AR-V7 with siRNA in 22Rv1 cells, which have abundant AR-V7 expression, re-sensitized the cells to enzalutamide. However, this was not the case in the bone metastasis-derived VCaP cell line, which also expresses AR-V7 but remains sensitive to ADT and enzalutamide [79]. Thus, it is likely that other AR variants play important roles in CRPC that need to be discovered.

One characteristic of PCa is that it has a long natural history, which therefore opens opportunities to investigate how signaling and metabolic molecules interact in promoting PCa growth and progression. Importantly, there is plenty of epidemiological and correlative data postulating that a Western diet (which is rich in long chain fatty acids) can promote PCa progression to lethal disease [14]. Currently, we do not know how the lipids themselves and the enzymes that metabolize them interact with AR and/or AR variants to promote CRPC.

## 4. Therapeutic Opportunities Targeting Lipid Metabolism and Androgen Signaling

Recent studies have indicated that lipids can fuel lethal prostate cancer in mouse models [23]. Although validation studies in human PCa remain to be done, these studies clearly place tumor lipids as potential drivers of aggressive PCa. Interestingly, AR signaling is known to be decreased in obesity [88], and obese patients with CRPC seem to have better overall survival [89]. These seemingly conflicting results clearly represent the difficulty in understanding how systemic lipid metabolism affects cancer outcomes in humans. However, it is worth noting that high-fat diets and weight gain are not equivalent, and the lipids delivered to or synthesized by the tumor might be the drivers of the association, not the systemic obesity per se. The specific use of the lipid is another angle to consider, since tumors with increased CPT1A expression may use the fatty acids for oxidation and promotion of growth and resistance to treatments (Figure 1).

The synthesis and use of lipids in PCa cells is regulated by androgens. Androgen receptor signaling in PCa is known to upregulate lipid biosynthetic enzymes such as FASN and ACACA (acetyl-CoA carboxylase alpha) [90,91]. On the other hand, androgens are known to increase palmitate oxidation in PCa cell lines [92,93]. What remains challenging is understanding how lipid biosynthetic and degrading enzymes interact with AR in the setting of CRPC, the incurable form of the disease. Current research is shedding light into this connection between AR and lipid metabolism, mainly in the form of novel pharmacological formulations or with drug-repurposing approaches. For example, a recent study has reported that blocking lipogenesis and AR signaling simultaneously strongly decreased PCa growth [43]. Zadra et al. showed that a novel, irreversible FASN inhibitor called IPI-9119 was able to change the cancer metabolome and induce PCa apoptosis. More interestingly, this drug-mediated inhibition of FASN also inhibited the expression of full-length AR and AR variants, including the clinically relevant AR-V7 variant. This AR downregulation was associated with endoplasmic reticulum stress (ER stress), likely mediated by dysregulation of the ER membrane lipid composition and phosphorylation of the translation initiation factor eiF2α, leading to a block in cellular protein synthesis [94]. Thus, these findings strongly connect the activity of lipogenesis in the cytoplasm with the AR expression in the nucleus of the cancer cell, opening the door for potential therapeutic opportunities to stop or ameliorate CRPC—for example, combining metabolic lipid inhibitors and anti-androgen drugs like abiraterone or enzalutamide, which are anti-androgens currently used in the clinic. The only downside of this potent FASN inhibition is the increased cholesterol synthesis and the biological implications this may have on tumor progression.

Other studies have found that 3-hydroxy-3-methyl-glutaryl-CoA reductase (HMGCR), a key enzyme in the cholesterol synthesis pathway, is elevated in enzalutamide-resistant PCa cells [95]. A combination of simvastatin and enzalutamide significantly inhibited the growth of enzalutamide-resistant PCa in vitro and in tumors, which appeared to be mediated by mTOR inhibition, blocking the synthesis of AR and enhancing the AR blockade with enzalutamide. Others have found that silencing the expression of a fatty acid elongase, ELOVL7, also leads to the regression of CRPC xenograft tumors in mice [96], highlighting the importance of AR-mediated lipid biosynthesis in CRPC progression. These studies represent another example of the therapeutic potential of lipid-lowering drugs to overcome endocrine resistance in CRPC. The connection of AR and lipid oxidation is equally important in the setting of PCa. Two proteomic analyses of primary [97] and bone metastasis [98] tumors have clearly shown that fat oxidation is an important pathway underlying the etiology and progression of PCa. Particularly, they found that bone metastasis could be classified into two distinct subgroups, one characterized by elevated expression of many canonical AR targets and by higher lipid oxidative function, and a second defined by low AR, high cell-cycle progression, and a more glycolytic metabolism. These metabolic subgroups could be used for patient stratification for suitable therapeutic options, perhaps combining metabolic and anti-AR therapies.

As mentioned above, a rational target for lipid oxidation in hormone-dependent cancer is CPT1. There are very few studies about the connection between CPT1A and AR in PCa [73,99]. Our work has shown that using the CPT1A inhibitor etomoxir in PCa cells results in decreased growth and AR expression [20], and this was associated with strong induction of ER stress and apoptotic ceramides production, as Loda’s group has recently found with FASN inhibitors [43] (see above). Further, when we targeted both FASN and CPT1 with orlistat and etomoxir, respectively, a stronger decrease in AR expression was observed, including AR variants, suggesting the possibility of a seemingly “futile” cycling of lipid synthesis and degradation necessary for AR-driven PCa growth and therapy resistance. Given this unexpected relationship between CPT1 and AR, we then tested combinations of drugs targeting fat oxidation (etomoxir, ranolazine, and perhexiline) and AR (enzalutamide or MDV3100) [73]. The effects of the combinations were very strong with the irreversible inhibitor etomoxir. However, results with the clinically approved drugs ranolazine and perhexiline were also significant, albeit less potent, suggesting a potential safe avenue to prolong the efficacy of the anti-androgen drugs with inhibitors targeting fat oxidation. The fates and implications of these combinations warrant further investigation with clinical trials.

Since pharmacological approaches always produce off-target effects, we also performed *CPT1A* genetic knockdowns in PCa cells. Interestingly, when we genetically decreased *CPT1A* expression in LNCaP cells, we observed less growth, a greater dependence on AR, and an increased sensitivity to the anti-androgen enzalutamide [73]. From a simplistic perspective, this increased dependence on AR does not seem to be a good approach, given that AR action is important for PCa cell survival. However, the greater dependence on AR facilitated the successful co-targeting of CPT1A and AR in decreasing PCa growth. Altogether, these results indicate that a reciprocal relationship may exist between the CPT1A enzyme in the mitochondria and the AR in the nucleus of the PCa (Figure 1). We believe this CPT1A–AR reciprocal interaction is critical for advanced PCa that has failed to respond to anti-androgen therapy. Other studies have also made a connection between AR and fat oxidation, using the chemo-preventive agent sulforaphane [100,101], although the effect does not appear to be mediated by direct inhibition of CPT1A since several enzymes of the beta-oxidation pathway were affected by the drug treatment. Thus, the intersection between two essential pathways in PCa, fat oxidation and androgen signaling, remain to be explored and exploited for targeted therapies. Ultimately, understanding how the tumor lipid availability, usage, and AR signaling relate to each other will help identify novel therapeutic and dietary targets for hormone dependent cancers.

## 5. Conclusions

Lipid metabolism rewires the PCa metabolome to support growth and resistance to endocrine therapies. It is still unknown which is first—the metabolic change or the AR reprogramming. It is possible that both occur in an interdependent manner, according to the tumor microenvironment and the nutrient availability. Future studies examining the lipid preference of tumors, from androgen-sensitive tumors to CRPC, will shed light on lifestyle and pharmacological interventions specific to the patient, enhancing the cancer treatment response and the quality of life of the patient. Exciting new research awaits.

## Figures and Tables

**Figure 1 ijms-20-02626-f001:**
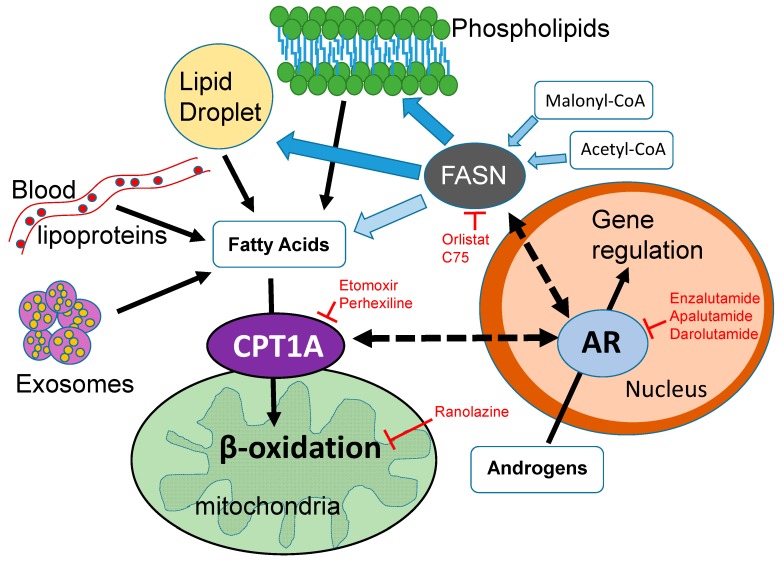
Cross-talk between lipid metabolism and the androgen receptor (AR) in the nucleus. The dotted arrows represent unknown mechanisms connecting fat burning in the mitochondria via CPT1A (carnitine palmitoyltransferase 1A) and fat synthesis in the cytoplasm via the FASN (fatty acid synthase) enzyme. Solid black arrows show direct connections. Red labels and red T-bars represent inhibitory drugs that are used in the clinic. Etomoxir and C75 are not used in humans. This diagram shows some of the sources of fatty acids available to the mitochondria, including the lipids that are newly synthesized via FASN. Light blue arrows show the substrates for FASN and its product, the fatty acid palmitate. This fatty acid of 16 carbons can be elongated and/or desaturated. The proportion of newly synthesized fatty acid that is burned in PCa is unknown, Dark blue arrows show that it can also be used for phospholipid and lipid droplet formation, which contains triglycerides and cholesterol esters. Exosomes delivering lipid droplets and phospholipids also represent another source of fatty acids for beta oxidation. Since androgens are known to regulate CPT1A and FASN enzyme activities, the coordination of fat synthesis and oxidation is likely modulated by the environmental context of the tumor, sometimes tipping the balance more towards synthesis and other times towards oxidation. Elucidating these tumor dependencies will increase the efficacy of lipid metabolic inhibitors and their combination with anti-androgen blockades.

**Table 1 ijms-20-02626-t001:** Representative studies of associations between dietary lipids and prostate cancer (PCa).

Lipids	Association Found	Reference
Dietary linoleic and alpha-linoleic acids	Metabolism at cellular level produces eicosanoids, some of which possibly function as tumor suppressors in PCa	Eicosanoids in prostate cancer. *Cancer Metastasis Rev.* **2018**. Review. [27].
Dietary alpha-linoleic acid (ALA)	Intake of ALA (mainly through mayonnaise consumption) associated with increased risk of lethal PCa prior to February 1994, when PSA testing began.	A 24-year prospective study of dietary α-linolenic acid and lethal prostate cancer. *Int. J. Cancer* **2018**. [28]
Saturated fats	High saturated fat intake associated with increased PCa aggressiveness	Saturated fat intake and prostate cancer aggressiveness: results from the population-based North Carolina-Louisiana Prostate Cancer Project. Prostate Cancer Prostatic Dis. **2017**. [29]
High-fat milk	High-fat milk intake associated with PCa progression (in localized PCa patients)	Dairy intake in relation to prostate cancer survival. Int J Cancer. **2017**. [30]
Omega 3 FA (fish-derived)	Higher Omega 3 intake may be associated with decreased PCa mortality	Fish-Derived Omega-3 Fatty Acids and Prostate Cancer: A Systematic Review. Integr Cancer Ther. **2017**. [31]
Monoethanolamine/lipid precursor	Anti-cancer activity of monoethanolamine evident and has clinic-use potential	Preclinical Development of a Nontoxic Oral Formulation of Monoethanolamine, a Lipid Precursor, for Prostate Cancer Treatment. Clin Cancer Res. **2017**. [32]
Cholesterol	Low cholesterol, low BMI, high physical activity associated with higher PCa risk **conflict with current PCa recommendations	Cholesterol and prostate cancer risk: a long-term prospective cohort study. BMC Cancer. **2016**. [33]
PUFA	Risk reductions observed for long-chain PUFA, short-chain PUFA, linoleic acid, and ALA in men under 62; increased risk for LA in men over 62.	Polyunsaturated fatty acids and prostate cancer risk: a Mendelian randomisation analysis from the PRACTICAL consortium. Br J Cancer. **2016**. [34].
Saturated fat, vegetable fat	For non-metastatic PCa, saturated fat intake may increase risk of death, while vegetable fat intake may lower it	Fat intake after prostate cancer diagnosis and mortality in the Physicians’ Health Study. Cancer Causes Control. **2015**. [35].
Fish and fish oils	Fish and fish oil consumption not consistently associated with reduction in PCa incidence, aggressiveness, and mortality	Systematic review of prostate cancer risk and association with consumption of fish and fish-oils: analysis of 495,321 participants. Int J Clin Pract. **2015**. [36].
Fried food	Larger intake of fried food associated with 35% increased risk of PCa	Fried food and prostate cancer risk: systematic review and meta-analysis. Int J Food Sci Nutr. **2015**. Review. [37].
Animal vs. vegetable fat	Potential benefit of vegetable fat for PCa-specific outcomes	Fat intake after diagnosis and risk of lethal prostate cancer and all-cause mortality. JAMA Intern Med. **2013**. [38].
SAFA, MUFA, PUFA	Balanced fat consumption diet may reduce the risk of PCa and prevent progression	Lipids and prostate cancer. Prostaglandins Other Lipid Mediat. **2012**. Review. [19]
Short chain fatty acids	High intake of total fat and certain saturated fatty acids may worsen PCa survival.	Dietary fatty acid intake and prostate cancer survival in Örebro County, Sweden. Am J Epidemiol. **2012**. [39].

Abbreviations: PCa; prostate cancer, FA; fatty acid, SAFA; saturated fats, MUFA; monounsaturated fats, PUFA; polyunsaturated fats, LA; linoleic acid, ALA; alpha-linoleic acid.

**Table 2 ijms-20-02626-t002:** Representative studies of associations between lipoproteins, exosomes, cholesterol delivery, and PCa.

Lipid(s) Investigated	Association Found	Reference
Cholesterol	CYP27A1 (PCa cellular cholesterol sensor) significantly contributes to PCa pathogenesis	CYP27A1 Loss Dysregulates Cholesterol Homeostasis in Prostate Cancer. Cancer Res. **2017**. [50]
Total cholesterol, LDL, HDL, and triglycerides	High LDL associated with longer recurrence-free survival	Prognostic Role of Preoperative Serum Lipid Levels in Patients Undergoing Radical Prostatectomy for Clinically Localized Prostate Cancer. Prostate. **2017**. [51].
Total cholesterol, LDL, HDL, and triglycerides	Upon irradiation with external beam therapy, LDL/HDL ratio in palliative subjects shows significant difference when compared to locoregional subjects	The Comparison and Estimation of the Prognostic Value of Lipid Profiles in Patients With Prostate Cancer Depends on Cancer Stage Advancement. Am J Mens Health. **2017**. [52].
Carnitine cycle (long-chain FA)	Carnitine cycle is indicated as a primary regulator of adaptive metabolic reprogramming in PCa cells	Deregulation of MicroRNAs mediated control of carnitine cycle in prostate cancer: molecular basis and pathophysiological consequences. Oncogene. **2017**. [53].
LDL and triglycerides	Weak evidence of higher LDL and triglycerides increasing PCa risk	Blood lipids and prostate cancer: a Mendelian randomization analysis. Cancer Med. **2016**. [54].
Serum triglyceride	Serum triglyceride levels not associated with PCa risk	Effects of Serum Triglycerides on Prostate Cancer and Breast Cancer Risk: A Meta-Analysis of Prospective Studies. Nutr Cancer. **2016**. [55].
Serum cholesterol, LDL	Total cholesterol and LDL correlated with PSA levels in cancer-free white males (prior to statin treatment)	Is PSA related to serum cholesterol and does the relationship differ between black and white men? Prostate. **2015**. [56].
Total cholesterol, LDL, and triglycerides	High cholesterol associated with increased risk lymph node metastasis; high LDL levels predict high Gleason scores	Serum lipid profiles and aggressive prostate cancer. Asian J Androl. **2015**. [57].
Total cholesterol, LDL, HDL, triglycerides	In dyslipidemia patients, elevated cholesterol associated with increased PCa recurrence, while elevated HDL associated with decreased PCa recurrence. Elevated triglycerides associated with increased PCa recurrence in general; elevated LDL not found to be associated with PCa recurrence	Serum lipid profile and risk of prostate cancer recurrence: Results from the SEARCH database. Cancer Epidemiol Biomarkers Prev. **2014**. [58].
Sphingolipids,, cholesterol, and phosphatidylserine	These lipids are enriched in the PC-3 exosomes and could be used as PCa biomarkers	Molecular lipidomics of exosomes released by PC-3 prostate cancer cells. Biochim Biophys Acta. **2013**. [59].
Phosphatidylserine, sphingolipids	The highest significance was shown for phosphatidylserine and lactosylceramide, which showed the highest patient-to-control ratio	Molecular lipid species in urinary exosomes as potential prostate cancer biomarkers. Eur J Cancer **2017**. [60].
Eicosanoids, fatty acids, and cholesterol	Exosomes are enriched in cholesterol and sphingomyelin and their accumulation in cells might modulate recipient cell homeostasis.	Exosomes as new vesicular lipid transporters involved in cell-cell communication and various pathophysiologies. Biochim Biophys Acta **2014**. [61]
Triglycerides	Exosomes from PCa cells enriched in triglycerides	Hypoxia induces triglycerides accumulation in prostate cancer cells and extracellular vesicles supporting growth and invasiveness following reoxygenation. Oncotarget **2015**. [62]

Abbreviations: PCa; prostate cancer, FA; fatty acid, LDL; low density lipoprotein, HDL; High density lipoprotein.

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
