# Peer review of "Lipid Metabolism and Endocrine Resistance in Prostate Cancer, and New Opportunities for Therapy"

_ijms, 2019, doi:10.3390/ijms20112626_

Round 1

Reviewer 1 Report

This is an excellent up to date review article with helpful summary tables of recent literature. The way cancer cells adapt to their environment to survive when the arterial supply is no longer sufficient and how diet may influence this adaptation is remarkable.. It is interesting that quality of diet but not obesity seems to be a major dietary factor in prostate cancer aggressive disease. Also of great interest that diabetes which is a major disease of disturbance of fat metabolism is not associated with increased aggressive prostate disease. A clearly written concise overview with excellent up to date references.

To the Editor.

A well written up to date review article worthy of publication.

Author Response

Responses to Reviewers Comments:

•           Reviewer 1

This is an excellent up to date review article with helpful summary tables of recent literature. The way cancer cells adapt to their environment to survive when the arterial supply is no longer sufficient and how diet may influence this adaptation is remarkable.. It is interesting that quality of diet but not obesity seems to be a major dietary factor in prostate cancer aggressive disease. Also of great interest that diabetes which is a major disease of disturbance of fat metabolism is not associated with increased aggressive prostate disease. A clearly written concise overview with excellent up to date references.

Thank you for the comments. We greatly appreciate the reviewer’s insights and time spent reading our manuscript.

Reviewer 2 Report

The manuscript by Drs Isabel Schlaepfer  and Gergana Stoykova is well written, well investigated and presents a good state of art of lipid metabolism in Prostate cancer (PC) with an interesting focus on the relation between lipid metabolism and androgen receptor (AR).

It will be interesting to understand clearly if there are differences between PC cells expressing AR and PC cells not expressing AR and lipid influence on PC progression, but I know that the scenario is not still richly investigated.

The English style is very good, tables are well performed and also the figure is very beautiful.

I have only some clarifications and/or questions:

1) In the introduction section the authors could add information about other early-stage treatments of Prostate cancer (PC) for having a larger audience (e.g prostatectomy, external beam radiotherapy and others). They could also add new techniques that are providing advances in management of PC and diagnosis, as new tracers PET/TC and novel focal therapies.

At this purpose, they could consider the following articles that can help the authors: i)doi:10.1007/s12020-019-01895-z; ii)doi:10.1021/acs.bioconjchem.5b00261

Furthermore, the should put on evidence in the introduction the modifications acquired by AR in PC progression or even that it can be lost by the cells. 

2) Another question is "Which is the role of tumor microenvironment (If it has a role) in lipid synthesis and uptake, or breaakdown and oxidation? The authors cite tumor microenvironment, but they not completely elucidate the role. In particular, have carcinoma-associated fibroblasts a role?

3) Are there in literature orgaonoid models showing data about androgen signalling and lipid metabolism?

4) Line 247: "The synthesis and use of lipids in PCa cells is regulated by androgens". A figure will be appreciated. 

5) Another question is: authors cite exosomes. The data in literature about exosomes and lipid in cancer are confined only to those reported in this review?

Author Response

•           Reviewer 2

The manuscript by Drs Isabel Schlaepfer  and Gergana Stoykova is well written, well investigated and presents a good state of art of lipid metabolism in Prostate cancer (PC) with an interesting focus on the relation between lipid metabolism and androgen receptor (AR).It will be interesting to understand clearly if there are differences between PC cells expressing AR and PC cells not expressing AR and lipid influence on PC progression, but I know that the scenario is not still richly investigated. The English style is very good, tables are well performed and also the figure is very beautiful. I have only some clarifications and/or questions:

1) In the introduction section the authors could add information about other early-stage treatments of Prostate cancer (PC) for having a larger audience (e.g prostatectomy, external beam radiotherapy and others). They could also add new techniques that are providing advances in management of PC and diagnosis, as new tracers PET/TC and novel focal therapies. At this purpose, they could consider the following articles that can help the authors: i)doi:10.1007/s12020-019-01895-z; ii)doi:10.1021/acs.bioconjchem.5b00261. Furthermore, the should put on evidence in the introduction the modifications acquired by AR in PC progression or even that it can be lost by the cells.

Thank you for the insightful comments. Although our manuscript is focused on therapies for advanced prostate cancer, we agree that the introduction needs more input about prostate cancer for a larger audience. We have extensively modified introduction to include localized disease and added a comprehensive current reference about diagnosis and treatment of prostate cancer (Litwin, JAMA 2017, ref #2). This reference also covers imaging modalities and new tracers. Regarding the new treatment modalities and the suggested articles, we have now included the suggested acs.bioconjchem.5b00261 reference in the second section of the manuscript, where we describe anti-AR approaches (lines 242-244, ref #86). Thank you for that.

2) Another question is "Which is the role of tumor microenvironment (If it has a role) in lipid synthesis and uptake, or breaakdown and oxidation? The authors cite tumor microenvironment, but they not completely elucidate the role. In particular, have carcinoma-associated fibroblasts a role?

Excellent point. We have now added a paragraph about cancer-associated fibroblasts as suppliers of energy-rich nutrients to cancer cells, lines 115-121. We also added two references (# 40 and 41).

3) Are there in literature orgaonoid models showing data about androgen signalling and lipid metabolism?

Organoids are becoming the preferred model for cancer studies. We agree with the reviewer that we needed to consider it. Interestingly, we only found two recent references focused on organoids, lipids androgen signaling. We have added these to the manuscript (references #42 and 43, line 121).

4) Line 247: "The synthesis and use of lipids in PCa cells is regulated by androgens". A figure will be appreciated.

Thanks for the suggestion. We have discussed this comment with our group, and we will likely work on another review article focused on molecular aspects of androgen-regulated metabolism. For the theme of the present review, we feel Figure 1 shows the connection between lipid synthesis and utilization and androgens. Lines 280 and beyond talk about this relationship.

5) Another question is: authors cite exosomes. The data in literature about exosomes and lipid in cancer are confined only to those reported in this review?

We agree. This was insufficient. We have now added three more references to Table 2 supporting the roles of exosomes in lipid delivery, including a recent study with lipids in human urinary exosomes (ref # 60).